# Systemic Antibiotics as an Adjunct to Subgingival Debridement: A Network Meta-Analysis

**DOI:** 10.3390/antibiotics11121716

**Published:** 2022-11-28

**Authors:** Ainol Haniza Kherul Anuwar, Roslan Saub, Syarida Hasnur Safii, Norintan Ab-Murat, Mohd Syukri Mohd Taib, Rokiah Mamikutty, Chiu Wan Ng

**Affiliations:** 1Department of Community Oral Health and Clinical Prevention, Faculty of Dentistry, Universiti Malaya, Kuala Lumpur 50603, Malaysia; 2Department of Restorative Dentistry, Faculty of Dentistry, Universiti Malaya, Kuala Lumpur 50603, Malaysia; 3Kinta District Office of Dental Health, Ipoh 30590, Malaysia; 4Department of Social and Preventive Medicine, Faculty of Medicine, Universiti Malaya, Kuala Lumpur 50603, Malaysia

**Keywords:** evidence-based dentistry, network meta-analysis, antibacterial agents, systemic antibiotics, adjunct, periodontal debridement, subgingival debridement, periodontitis

## Abstract

This review aimed to evaluate the effectiveness of systemic antibiotics as adjunctive treatment to subgingival debridement in patients with periodontitis. Randomized controlled trials were included that assessed the effectiveness of systemic antibiotics in improving periodontal status, indicated by clinical attachment gain level, probable pocket depth reduction, and bleeding on probing reduction of patients with any form of periodontitis at any follow-up time. Network meta-analyses with a frequentist model using random effects was employed to synthesize the data. The relative effects were reported as mean difference with a 95% confidence interval. Subsequently, all treatments were ranked based on their P-scores. A total of 30 randomized controlled trials were included in this network meta-analyses. Minimally important clinical differences were observed following the adjunctive use of satranidazole, metronidazole, and clindamycin for clinical attachment gain level and probable pocket depth reduction. For bleeding on probing reduction, minimally important clinical differences were observed following the adjunctive use of metronidazole and a combination of amoxycillin and metronidazole. However, the network estimates were supported by evidence with certainty ranging from very low to high. Therefore, the findings of this network meta-analyses should be interpreted with caution. Moreover, the use of these antibiotics adjunct to subgingival debridement should be weighed against possible harm to avoid overuse and inappropriate use of these antibiotics in patients with periodontitis.

## 1. Introduction

From 1990 to 2010, severe periodontitis was identified as the sixth most prevalent condition affecting 538 million (7.4%) of the world’s adult population [1]. Unfortunately, although periodontal diseases are largely preventable, easily diagnosed, and effectively managed [2], the prevalence of severe periodontitis has continued to increase by 8.44% over the past three decades (1990–2019), affecting 1.1 billion adults globally [3]. Periodontitis shares common modifiable risk factors and social determinants with other major chronic non-communicable diseases such as heart disease, diabetes, and hypertension [4,5]. Among the risk factors, tobacco smoking, obesity, poor nutrition, and physical inactivity are associated with an increased risk of periodontitis [6]. If left untreated, this disease causes mobility of the teeth, and ultimately tooth loss [7,8]. Subsequently, the affected individuals may also have their nutrition affected, experience poor quality of life, and suffer low self-esteem [2,9,10,11,12].

To treat periodontitis, the removal of the causative factor (i.e., dental biofilm) from the tooth surface is imperative. Initially, all patients are treated with non-surgical periodontal therapy where the diseased sites are treated with subgingival debridement using manual or ultrasonic instruments, individually or in combination, to eliminate the calculus which acts as a plaque-retentive factor [13,14,15,16]. However, in some patients, subgingival debridement alone may not produce desirable clinical outcomes, for several potential reasons including the presence of inaccessible sites and colonization of tissue-invading bacteria [13]. For such patients, the European Federation of Periodontology clinical practice guidelines recommended the adjunctive use of systemic antibiotics in the treatment of periodontitis [16]. Nevertheless, the authors only recommended adjunctive use of systemic antibiotics for specific patient categories (e.g., young adult patients with generalized Stage III periodontitis).

In dentistry, in addition to their adjunctive use in the treatment of periodontitis, antibiotics are widely prescribed prophylactically and therapeutically [17,18]. Concurrently, antibiotics are also applied to prevent and treat a wide range of diseases [19]. Not limited to humans, antibiotics are also broadly used in veterinary medicine and agriculture [20,21]. It is of great concern that these mutual and unselective uses of antibiotics have contributed to the rise of antibiotic resistance [22,23]. The issues of antimicrobial drug resistance have caused serious socioeconomic and health problems worldwide [24].

In addition to contributing to increased antimicrobial resistance, the mutual use of systemic antibiotics has raised concerns about their impact on patients and public health [16,17,25]. They are associated with adverse events such as allergic reactions, gastrointestinal, cardiac, renal, and neurological effects [26]. Considering these problems, it is important to avoid unnecessary adjunctive use of antibiotics. With regards to NSPT, several meta-analyses have been conducted to determine the most effective and safest use of systemic antibiotics adjunctive to subgingival debridement in the treatment of patients with periodontitis [25,27,28].

There have been mixed findings regarding the effectiveness of the adjunctive use of systemic antibiotics in non-surgical periodontal therapy for patients with periodontitis. Meta-analysis by Teughels et al. [29] concluded that the adjunctive use of systemic antibiotics, namely a combination of amoxycillin and metronidazole (AMOX+MET), metronidazole (MET) alone, and azithromycin (AZ), can be effective in improving the periodontal status of patients with periodontitis. Meanwhile, a network meta-analyses by Sgolastra et al. [28] concluded that AMOX+MET is the best adjunct for patients with periodontitis. However, Khattri et al. [27] found inconclusive evidence to reach such a conclusion. Despite these findings, it is important to note that some of the antibiotics in the included studies were used in combination with other active adjuncts such as chlorhexidine that may confound the true effect of the adjunctive use of these antibiotics. Additionally, they also included primary studies that assessed discontinued antibiotics and others that are not commercially available. Hence, there remains a need to determine the true effect of commercially available systemic antibiotics adjunctive to subgingival debridement for improving the periodontal status of patients with periodontitis. This study provides information on the superiority of these systemic antibiotics as adjuncts in non-surgical periodontal treatment, to facilitate informed decision-making by clinicians and decision-makers.

## 2. Materials and Methods

### 2.1. Review Protocol and Registration

The network meta-analysis (NMA) was conducted and reported according to the PRISMA (preferred reporting items for systematic review and meta-analyses) standard for network meta-analyses [30] and the *Cochrane Handbook for Systematic Reviews of Interventions* [31]. The protocol of this NMA was registered with the International Prospective Register of Systematic Reviews (PROSPERO) (Registration number: CRD42021252397) and obtained approval from the Medical Ethics Committee, Faculty of Dentistry, Universiti Malaya [Reference number: DF CO2015/0089 (P)].

### 2.2. Eligibility Criteria

#### 2.2.1. Type of Study

Only randomized controlled trials (RCTs) (including pilot studies) conducted on humans, available as full text documents, written in English, and published from the year 2000 onwards were included.

#### 2.2.2. Type of Population

All patients with any form of periodontitis as defined by either the 1999 [32] or 2017 [33] classifications were included. Studies that specifically aimed at assessing smokers, patients with systemic diseases or special conditions, or patients harboring specific types of periodontal pathogens were excluded.

#### 2.2.3. Type of Interventions

Relevant interventions included any systemic antibiotics used adjunctively to subgingival debridement in the non-surgical periodontal treatment of patients with periodontitis, initiated or prescribed during the cause-related therapy. Systemic antibiotics used in combination with other adjuncts (e.g., chlorhexidine), or that were discontinued or not commercially available were excluded. The included intervention studies used subgingival debridement with or without placebo as comparator. Studies using other type of comparators were excluded.

#### 2.2.4. Type of Outcomes

Studies were included if they reported the following outcomes: (i) pre- and post-intervention full-mouth clinical attachment level (CAL), probable pocket depth (PPD), and bleeding on probing (BOP), along with their standard deviations (SDs); (ii) mean difference (MD) in CAL, PPD, and BOP between baseline and follow-up visits; and/or (iii) adverse events associated with the adjunctive use of the antibiotics. Outcomes reported at short-term (≤3 months), intermediate (>3 to <12 months), and long-term (≥12 months) follow-up periods were included.

### 2.3. Information Sources

Eight electronic databases were searched, namely, PubMed, MEDLINE (via EBSCOhost), Cochrane Library, Web of Science, SCOPUS, CINAHL (via EBSCOhost), Dentistry & Oral Sciences Source (via EBSCOhost), and Index Medicus for the South-East Asian Region (IMSEAR). In addition to the electronic databases, the reference lists of selected studies and review articles were also screened to identify relevant papers.

### 2.4. Search

The terms used for the search were identified through scientific databases (i.e., PubMed and MEDLINE), published papers, and consultation with clinical experts. Alternative spellings (including both US and British English) were also considered (Appendix A). Prior to searching the electronic databases, this search strategy was reviewed and validated by a senior medical librarian (R.H.S) to confirm its appropriateness for this review. The search was conducted using commands specific to each database interface. If the volume of retrieved records was very high, the search was limited to RCTs published between 2000–2021 in the English language. The retrieved records were exported to EndNote version X9, where duplicates were identified and removed. All searches were updated and concluded in September 2021.

### 2.5. Study Selection

The study selection was performed using a two-stage selection process. Firstly, all retrieved titles and abstracts were screened, then the full texts of the articles selected during the first stage were assessed against the eligibility criteria prior to their inclusion in the NMA. To minimize selection bias, parallel independent assessments were carried out by two reviewers (A.H. and M.S.). Inter-rater kappa scores were calculated to indicate the reliability of the decisions made by the reviewers. Any disagreements between reviewers were discussed and resolved. If required, a third reviewer was consulted to resolve any disagreements between the two reviewers.

### 2.6. Data Extraction

Relevant information regarding the general characteristics (i.e., author, year, country, funding, trial design, population, age, gender, and smoking status), intervention characteristics (i.e., delivery of NSPT, test group, control group, and treatment provider), and reported outcomes and measurement methods (i.e., CAL gain, PPD reduction, BOP reduction, and adverse events) were extracted from the included studies, along with the findings reported. A standardized data-extraction form was developed and utilized to ensure consistency and reduce bias, as well as to improve the validity and reliability of this review. The pilot testing for data extraction was performed on 10% of the included studies. Any ambiguities or amendments made to the data-extraction forms were recorded. Ideally, data extraction should be performed by two reviewers independently. However due to time constraints, as an accepted minimum [34], the first reviewer (A.H.) extracted the data, and the second reviewer (M.S.) independently checked the data-extraction forms for accuracy and completeness. Disagreements between reviewers were resolved through discussion or involvement of the third reviewer, if necessary.

### 2.7. Geometry of the Network

A network plot was constructed for each outcome according to the follow-up period (i.e., short-term, intermediate, or long-term) to ensure transitivity of the treatment network. The size of the node (circle) denotes the number of participants involved, and the size of the connection (line thickness) represents the number of studies per treatment. The line of the network plot represents the direct comparison between the interventions.

### 2.8. Risk of Bias within Individual Studies

The risk of bias was assessed based on available data as reported in the full text articles of the included studies. Two independent reviewers (A.H. and R.M.) performed this assessment using the revised Cochrane risk of bias tool for randomized trials (RoB 2) [35]. Inter-rater agreement was calculated to assess the reliability of the assessments made by the reviewer. The summary, as well as the plots for risk of bias assessment, were generated using the risk-of-bias visualization (robvis) tool [36].

### 2.9. Data Synthesis

The findings were analyzed both narratively and quantitatively. The outcomes reported by the authors were transformed into common outcome statistics for each study. As the outcomes of interest were MD and SD, the outcomes presented in the median range or median-quartile range were transformed using methods proposed by Luo et al. [37], Shi et al. [38,39], and Wan et al. [40]. Prior to conducting NMA, pairwise meta-analyses using a random effect model were carried out to evaluate the between-study heterogeneity of all directly compared interventions, using *dmetar* and *meta* packages [41] in RStudio (version 1.4.1106) [42]. Subsequently, NMA was applied to evaluate the relative effect of the included treatment. Subgingival debridement alone or with placebo was used as a reference intervention to ensure that the transitivity of the analysis was not violated. NMA using a frequentist model was conducted using the *netmeta* package [43]. A random-effect model was employed to assess the presence of between-study variability across the network. The effect estimates of all direct and indirect comparisons were presented in league tables. All treatments were ranked based on their P-score, with treatments with higher P-scores ranking better than the other competing treatments [44]. Statistical heterogeneity was tested using the Tau^2^ test, as well as I^2^ statistics [31]. The minimally important clinical difference (MICD) following treatment was set at 1 mm for CAL gain and PPD reduction and 5% for BOP reduction, as suggested by Khattri et al. [27].

### 2.10. Assessment of Inconsistency

Inconsistency in the NMA was assessed by contrasting a direct estimate with an indirect estimate via separate indirect from direct evidence (SIDE), using a back-calculation method in the *netmeta* package [43] to determine the statistical agreement between the direct and indirect comparisons [45,46,47].

### 2.11. Subgroup Analyses

Subgroup analyses were conducted to explore potential reasons for differences, by splitting the studies into less heterogenous groups according to the following: (i) type of periodontitis (as reported in the included studies); (ii) smoking status (i.e., including or excluding smokers); (iii) antibiotics initiation (i.e., after first subgingival debridement, after completion of subgingival debridement, before subgingival debridement, or unclear); and (iv) antibiotic dosage (as reported in the included studies). To ensure sufficient statistical power to detect subgroup differences, these analyses were conducted only for outcomes recorded in at least ten included studies.

### 2.12. Sensitivity Analyses

Sensitivity analyses were carried out by re-running the NMA after excluding studies with parameters that may affect the estimates of treatment effect, as follows: (i) high risk of bias; (ii) outlier or influential cases; (iii) imputed or transformed data.

### 2.13. Assessment of Publication Bias

The potential for publication bias for each outcome was explored using funnel plots and Egger’s test, if there were more than ten available studies.

### 2.14. Assessment of Body of Evidence Sertainty

The grades of recommendation, assessment, development, and evaluation (GRADE) approach to assess the certainty of the body of evidence was employed for rating the quality of treatment-effect estimates provided by NMA [48]. The RCTs were initially set as high quality [48,49,50]. For direct and indirect estimates, the evidence was downgraded in the presence of serious risk of bias, inconsistency, indirectness, imprecision, or publication bias [48]. The direct and indirect estimates contributed to the ratings of the network estimates. These ratings were further downgraded in the presence of serious incoherence and imprecision [48,51]. An interpretive overview of the findings is presented in the “summary of findings” table.

## 3. Results

### 3.1. Study Selection

Of 2953 studies identified through databases, only 87 full texts were assessed against the eligibility criteria. Of these, 30 studies fulfilled the eligibility criteria and were included in the NMA (Figure 1). Information about the excluded studies and the reasons for their exclusion is presented in Appendix A. The inter-rater kappa score for title and abstract screening was 0.97 and for the full text screening was 0.90, both indicating an almost perfect agreement between reviewers [52].

### 3.2. Study Characteristics

#### 3.2.1. General Characteristics

The included studies were published between 2001–2021. These studies were conducted in Asia [53,54,55,56,57,58,59,60], South America [61,62,63,64,65,66,67,68,69,70,71,72], Europe [73,74,75,76,77,78,79,80], and Australia [81,82]. They were mostly single-centered [53,54,55,56,57,58,59,60,61,62,64,65,66,67,68,69,70,71,72,73,74,75,78,80,81,82], used a parallel-group approach [53,54,55,56,57,60,61,62,63,64,65,66,67,68,69,71,72,73,74,75,76,77,78,79,80,81,82], and were funded by government agencies [61,64,65,66,67,68,69,70,71,72,74,76,77]. These studies defined their participants as having chronic [53,56,57,59,60,65,68,70,71,78,79], aggressive [54,61,62,63,64,67,72,73,75,80], severe or advanced [55,58,69], moderate-to-advanced [74,76,81,82], Stage III and Stage IV [77], or mild-to-moderate [66] periodontitis. Twelve studies included smokers as their subjects [54,56,66,67,68,70,71,74,76,77,79,80]. The mean age of the subjects ranged from 20.1 to 58.5 years old. The general characteristics of the included studies according to the adjuncts used are summarized in Appendix A.

#### 3.2.2. Intervention Characteristics

Oral hygiene instruction was given to the subjects as part of the treatment, except in five studies [57,60,66,73,75]. Subgingival debridement was provided in one stage [55,58,61,62,63,69,70,71,72,73,74,75,76] or multiple stage [53,54,56,67,68,70,71,77,78,79,80,81,82]. Fonseca et al. [66] used both approaches in their study. In the study by Čuk et al. [74] only sites with PD ≥5 mm received subgingival debridement, which was performed in two sessions. The remaining studies failed to describe clearly the delivery of their interventions [57,59,60,65,72,73]. The treatments were mainly provided by periodontists, dentists, and dental hygienists [55,56,61,62,63,64,65,66,67,68,69,70,71,73,74,78,81,82]. Antibiotics that were used as adjuncts in the included studies were AMOX+MET [58,62,65,70,71,72,73,75,76,81,82], AZ [54,55,56,57,66,67,68,70,74,77,81,82], MET [71,78,80], clarithromycin (CLM) [60,61,64], moxifloxacin (MOX) [62,63], doxycycline (DOX) [78,82], cefixime (CEF) [75], clindamycin [78], minocycline (MINO) [53], satranidazole (SZ) [59], and secnidazole (SEC) [57]. Most studies initiated antibiotics immediately after completion of subgingival debridement [54,56,58,59,64,65,68,70,72,74,75,76,77,78,79,80]. Various dosages of antibiotics were prescribed in these studies. Appendix A summarizes the intervention characteristics of the included studies.

#### 3.2.3. Reported Outcome and the Outcome Measurements

All studies reported PPD reduction following intervention. Only one did not report CAL gain [53], and ten studies did not report BOP reduction [56,57,59,60,66,72,76,78,79,80]. In terms of the safety of using systemic antibiotics, six RCTs did not report adverse events [53,57,60,66,73,78]. Most studies had one examiner to measure the outcomes, except for eight studies [55,60,66,70,71,75,76,78]. The examiners in 15 studies were calibrated and blinded [54,56,59,61,62,63,64,65,67,69,72,73,77,81,82]. Remaining studies did not provide clear descriptions of their examiners [55,58,70,75]. The majority used only manual periodontal probes to measure outcomes [53,54,56,57,59,60,61,62,63,64,65,66,67,68,69,70,71,72,74,75,77,78,79,80,81,82]. However, only four studies provided clear operational definitions of all the outcome measurements [57,59,69,70]. The reported outcomes and the outcome measurements of the included studies are summarized in Appendix A.

### 3.3. Risk of Bias

Ten studies were at low risk [58,61,62,63,64,65,68,70,71,73], three at unclear risk [53,60,78], and 17 at high risk [54,55,56,57,59,66,67,69,72,74,75,76,77,79,80,81,82] of overall bias (Figure 2). Deviations from the intended interventions contributed to the risk of bias, wherein about half of the studies performed per-protocol instead of intention-to-treat analysis. The characteristics of subjects that were excluded from the analysis were not clearly described. The authors’ judgement and support for judgements for each risk of bias in all included studies are summarized in Appendix A. There was no disagreement between the two reviewers during the risk-of-bias assessment (inter-rater agreement: 100%).

### 3.4. Network Meta-Analyses

The outcomes in the included studies were reported at diverse follow-up periods. Therefore, in this NMA the findings are reported according to follow-up periods: (i) Short-term (≤3 months); (ii) intermediate (3 months to <12 months); and (iii) long-term (≥12 months). The pairwise comparison and sources of heterogeneity (i.e., outliers and influential cases) are presented in Appendix A. The league tables including the direct and indirect comparisons in the NMA of the treatments for all outcomes are shown in Appendix A.

#### 3.4.1. CAL Gain

The network and forest plots of the NMA for CAL gain following adjunctive used of systemic antibiotics are illustrated in Figure 3. In terms of short-term CAL gain, out of 11 included treatments, adjunctive use of SZ, MOX, and AMOX+MET showed significant CAL gain when compared with subgingival debridement alone. SZ was the most superior with 100% probability of being ranked first when all treatments including the control group were compared, followed by MOX and AMOX+MET. Meanwhile, only SZ showed minimally important clinical difference (MICD). It is important to note that substantial between-studies heterogeneity (I^2^ = 78%) was identified in this head-to-head comparison.

Referring to intermediate-term CAL gain, out of 10 included treatments, adjunctive use of SZ was again superior with 100% probability when all treatments were compared, followed by MET, clindamycin, MOX, and AMOX+MET. Statistically, these antibiotics showed significant CAL improvement but only SZ showed MICD. However, this comparison also had substantial (I^2^ = 81.5%) between-studies heterogeneity.

Regarding long-term CAL gain, only six treatments were included in the NMA. The most superior antibiotic in this comparison was MET, with 99% probability of being ranked first when all treatments were compared, followed by clindamycin, DOX, and AMOX+MET. These antibiotics showed significant CAL gain when compared with subgingival debridement alone. MICD was only obtained with the adjunctive use of MET and clindamycin. Substantial between-studies heterogeneity (I^2^ = 67.5%) was also observed in this comparison.

#### 3.4.2. PPD Reduction

The network and forest plots of the NMA for PPD reduction following adjunctive use of systemic antibiotics are illustrated in Figure 4. Out of 11 treatments, the adjunctive use of SZ was the most effective, with 99% probability of being ranked first, followed by MET, MOX, and AMOX+MET when all treatments including the control group were compared in terms of short-term PPD reduction. The adjunctive use of these antibiotics showed significance difference statistically, but only SZ showed MICD. However, the between-studies heterogeneity was very high with I^2^ of 97%.

For PPD reduction in the intermediate term, out of ten treatments, SZ (P-score: 1.00) was again superior to other antibiotics followed by MET, clindamycin, AMOX+MET, and MOX. Although these antibiotics were associated with significant PPD reduction levels, only SZ showed clinical significance. The between-studies heterogeneity was also high (I^2^ = 89.4%) for this comparison.

Out of six treatments included in the NMA for long-term PPD reduction, MET ranked first with 99% probability of being the most effective. MET, clindamycin, DOX, and AMOX+MET showed significant differences when compared with subgingival debridement alone. Clinically, both MET and clindamycin showed MICD following their adjunctive used. The between-studies heterogeneity was substantial with I^2^ of 72.8%.

#### 3.4.3. BOP Reduction

The network and forest plots of the NMA for BOP reduction following adjunctive use of systemic antibiotics are shown in Figure 5. Regarding short-term BOP reduction, only two adjunctive antibiotics showed both statistical and clinical significance when compared with subgingival debridement alone, namely MET and AMOX+MET. MET ranked first (P-score: 1.00) followed by AMOX+MET (P-score: 0.76) when the six treatments (including NSPT alone) were compared. There was considerable (I^2^ = 90.3%) between-studies heterogeneity in this head-to-head comparison.

For intermediate-term outcomes, adjunctive use of AMOX+MET was the most effective with 96% probability of being ranked first when all five competing treatments were compared. However, the I^2^ for this comparison was 92.1% which indicated considerable between-studies heterogeneity.

In the long-term results, only AMOX+MET (P-score: 1.00) showed significant BOP reduction and MICD when compared with subgingival debridement alone. Only three treatments were included in this analysis, with substantial (I^2^ = 57.4%) between-studies heterogeneity.

#### 3.4.4. Adverse Events

There were no serious adverse events reported in the included studies, except for in a study conducted by Harks et al. [77] where allergic and anaphylactic reaction was reported by subjects in the test group following the adjunctive used of AMOX+MET (Appendix A). In the same study, subjects in the control group also reported allergic reaction after receiving the treatment.

### 3.5. Assessment of Inconsistency

The inconsistency assessment was only performed for short- and intermediate-term outcomes, because the comparisons in the long-term outcomes contained only direct or indirect estimates. For short-term CAL gain, inconsistencies were found in three comparisons: (i) AMOX+MET vs. MET (*p* = 0.01); (ii) MET vs. NSPT alone (*p* = 0.01); and (iii) MOX vs. NSPT alone (*p* = 0.04), whereas for intermediate-term CAL gain, inconsistency was not observed in any of the comparisons. Regarding PPD reduction and BOP reduction, none of the comparisons of either the short- or intermediate-term outcomes showed inconsistency between their direct and indirect estimates (Appendix A).

### 3.6. Subgroup Analyses

Subgroup analyses were conducted only for outcomes with at least ten included studies. Therefore, this analysis was performed for CAL gain at all follow-up periods (Appendix A), PPD reduction at all follow-up periods (Appendix A), and short- and intermediate-term BOP reduction (Appendix A). Among the study characteristics, the following were explored to identify potential sources of heterogeneity: (a) type of periodontitis; (b) smoking status; (c) antibiotics initiation; and (d) dosage.

In terms of CAL gain outcomes, significant subgroup differences were observed in all subgroup analyses (*p* < 0.05), except for smoker status (*p* = 0.07) and antibiotic initiation (*p* = 0.53) for long-term CAL gain. For PPD reduction at all follow-up periods, significant subgroup effects were detected only for the type of periodontitis (*p* < 0.05) and the dosage (*p* < 0.0001). Concerning short- and intermediate-term BOP reduction, significant subgroup differences (*p* < 0.05) were detected in all subgroup analyses except for smoking status with *p*-values of 0.27 and 0.67, respectively.

### 3.7. Sensitivity Analyses

#### 3.7.1. CAL Gain

The sensitivity analyses for short-term CAL gain (Appendix A) showed that the results of the primary analysis remain robust, especially for the top six treatments which remained superior when all treatments were compared. However, it is important to note that when studies at high risk of bias were removed, SZ was excluded and the superiority of AZ and MET diminished in comparison with subgingival debridement alone.

Conversely, the robustness of findings for intermediate-term CAL gain (Appendix A) was applicable only for the four most effective antibiotics namely SZ, MET, clindamycin, and MOX. Nonetheless, SZ were not included in the analysis when the studies at high risk of bias were excluded. AMOX+MET was found to be inferior to DOX after removal of studies with outliers and influential cases, as well as those at high risk of bias. Like short-term CAL gain, the effectiveness of AZ was reduced compared with the primary analysis. As for long-term CAL gain, the primary result remained robust (Appendix A).

#### 3.7.2. PPD Reduction

The primary findings of the short-term outcome (Appendix A) remained robust when studies with imputed data and high risk of bias were removed. However, when the outliers and influential cases were removed, CLM became less superior compared with the primary analysis, and DOX was found to be less effective than NSPT alone.

For intermediate-term outcomes (Appendix A), the findings remained robust when the outliers and influential cases were removed, but AZ was found to be inferior to NSPT alone when studies with high risk of bias were removed. Likewise, for the long-term PPD reduction (Appendix A), the sensitivity outcome was similar to the primary analysis, while AZ became inferior to NSPT alone in the sensitivity analysis.

#### 3.7.3. BOP Reduction

As shown in Appendix A, the findings of the sensitivity analysis were found to be comparable to the primary analysis. However, MOX became inferior to CLM when the outliers and influential cases were omitted. Otherwise, the findings for intermediate- (Appendix A) and long-term (Appendix A) BOP reduction remained robust.

### 3.8. Publication Bias

This assessment was carried out only for outcomes supported by at least ten studies. For CAL gain (Appendix A), publication bias was only detected in relation to long-term outcome (*p* < 0.01), while for PPD reduction (Appendix A), publication bias was observed in the intermediate- (*p* = 0.05) and long-term (*p* < 0.01) outcomes. Nevertheless, the studies outside the funnel plots were also outliers or influential cases which were removed during the sensitivity analyses to ensure the robustness of the primary analysis. With regards to BOP reduction (Appendix A), no publication bias was observed (*p* > 0.05).

### 3.9. Summary and Certainty of Evidence

The findings for all outcomes (CAL gain, PPD reduction, and BOP reduction) for adjunctive use of systemic antibiotics, along with their certainty of evidence, are illustrated according to follow-up period in Figure 6, Figure 7, Figure 8, Figure 9, Figure 10, Figure 11, Figure 12, Figure 13 and Figure 14. The certainty of evidence ranged from high to very low (Appendix A). In this NMA, evidence certainty was downgraded mainly due to serious risk of bias, inconsistency, indirectness, or imprecision. Therefore, the NMA estimates presented here should be interpreted with caution.

## 4. Discussion

### 4.1. Summary of Findings

Applying the evidence-based principle, NMA was conducted to determine the relative effect of systemic antibiotics as adjuncts to subgingival debridement in the non-surgical periodontal treatment of patients with periodontitis. This study was carried out not only to ensure that treatments provided by clinicians are effective but also to improve the quality of patient care [83]. By integrating their own experience and skill with the findings of this present NMA and with other published studies or guidelines, clinicians will be able to make informed clinical decisions and to educate their patients in making treatment choices.

To summarize the findings of the NMA, MICD was used as defined in a review by Khattri et al. [27]. Considering there had been no established cut-off value to show the clinical significance of an intervention in improving periodontal status, the cut-off values determined by Khattri et al. [27] can help clinicians to understand and value these research findings. It is important for researchers to report clinical significance, in order for clinicians to adhere to the principles of evidence-based practice. By reporting clinical significance data, evidence can be disseminated effectively in ways that are more comprehensible for the end-users (e.g., policy- or decision-makers, healthcare providers, and patients) [84,85]. Moreover, using only the systemic antibiotics that resulted in MICD following their adjunctive use in non-surgical periodontal treatment may prevent inappropriate and unselective use of antibiotics.

For CAL gain and PPD reduction, MICD was only observed following the adjunctive use of SZ for short- and intermediate-term outcomes, whereas MICD for long-term CAL gain was observed following the adjunctive use of MET and clindamycin. However, the effectiveness of SZ as an adjunct to subgingival debridement was supported by low-certainty evidence due to very serious risk of bias. It should also be noted that SZ is not commercially available in all countries and is used to treat many health disorders (e.g., amebic liver abscess) [86]. The use of SZ is associated with side effects such as headaches, dry mouth, weakness, and dizziness [87]. Thus, future well-designed and high-quality RCTs are required to report the effectiveness of SZ to support its use in the treatment of patients with periodontitis. For long-term outcomes, supported by evidence with moderate certainty, MICD was observed following the adjunctive use of MET and clindamycin. Hence, the adjunctive use of both antibiotics may be considered to improve the CAL and PPD of patients.

With regards to BOP reduction, likewise supported by evidence with moderate certainty, the adjunctive use of MET showed MICD post-intervention for short-term outcomes. Meanwhile, the adjunctive use of AMOX+MET showed MICD at all follow-up periods but the findings for short- and intermediate-term outcomes were supported by evidence with very low and low levels of certainty, respectively. Nonetheless, high-certainty evidence supported the adjunctive use of AMOX+MET for long-term BOP reduction. Therefore, the use of MET and AMOX+MET as adjuncts to subgingival debridement may also be considered in the treatment of patients with periodontitis.

Subgroup analyses were conducted to explore potential sources of heterogeneity between studies. For all the parameters explored, the findings revealed that the type of periodontitis and the antibiotic dosages may have contributed to the differences. Regarding the type of periodontitis, recent guidelines recommend prescribing systemic antibiotics as an adjunct to subgingival debridement only for patients in specific categories (e.g., young adults with generalized stage III periodontitis) [16]. In this present NMA, MICD was observed in the PPD reduction of patients with severe or advanced periodontitis, and BOP reduction in cases of chronic and severe or advanced periodontitis following the adjunctive use of systemic antibiotics. This finding is somewhat in line with the recommendations that systemic antibiotics should only be prescribed for severe forms of periodontitis if necessary, and routine use of antibiotics should otherwise be avoided. However, the findings should be interpreted with caution as they are supported evidence ranging from very low to high certainty. Furthermore, diverse definitions of periodontitis were used in the published studies. Therefore, more high-quality RCTs using a standardized definition based on current classification [33] should be conducted to inform the effectiveness of systemic antibiotics in patients with different type of periodontitis. Similarly, a variety of antibiotic dosages were prescribed to treat these patients, which mostly did not result in MICD following intervention. Hence, there is a need to identify the most appropriate doses to avoid inappropriate use of these antibiotics.

However, the marginal additional clinical benefit observed following the adjunctive use of these antibiotics should be weighed against possible severe risks (e.g., microbiological side effects and an increase in bacterial resistance) [22,23,88]. Since antibiotics have been widely prescribed prophylactically and therapeutically in dentistry [89], restrictive action should be taken to avoid overuse and inappropriate use of these antibiotics for treatment of patients with periodontitis. Among all regions globally, qualitative risk assessment of antibiotic resistance has shown that Southeast Asia is probably at the highest risk globally for the emergence and spread of antimicrobial resistance [89]. Hence, prescription of these antibiotics should be limited as much as possible and only used as a last resort to treat patients with periodontitis [22]. To treat patients with periodontitis, the effect of proper mechanical debridement along with modification of risky behaviors should not be underestimated nor disregarded [22]. If clinicians continue to prescribe antibiotics uncontrollably, the healthcare system may be exposed to unnecessary expenditure resulting in severe economic impact [90,91].

### 4.2. Comparisons with Previous NMA

According to previous NMA [24,87], the adjunctive use of AMOX+MET showed improved clinical outcomes compared with other systemic antibiotics, namely MET alone, AZ, MOX, and DOX. Additionally, the adjunctive use of AMOX+MET is recommended in the most recent European Federation of Periodontology clinical practice guidelines [16]. Previous meta-analysis by Teughels et al. [29] informed this recommendation, in which the authors reported significant clinical improvement following the use of AMOX+MET as an adjunct to subgingival debridement.

In contrast, the findings of the present study found that AMOX+MET was superior only for intermediate- and long-term BOP reduction. In several previous primary studies [24,29,87], AMOX+MET was prescribed in combination with chlorhexidine. Because such studies were excluded from the present NMA, it can be concluded that the superiority exhibited by AMOX+MET in the previous NMA may have been affected by the confounding effect of chlorhexidine [92]. However, as the certainty of evidence in the present NMA ranged from very low to high, future high-quality RCTs are necessary in order to reach a strong conclusion regarding the most effective systemic antibiotics adjunctive to subgingival debridement in non-surgical periodontal therapy for patients with periodontitis.

Furthermore, previous NMA included only patients with chronic periodontitis [28] or aggressive periodontitis [93], whereas in this study, patients diagnosed as having both chronic and aggressive periodontitis were included. This is in line with the current classification of periodontitis, where conditions previously defined as chronic and aggressive periodontitis are grouped into a single category of periodontitis [94]. The other two categories of periodontitis in the current classification are necrotizing periodontitis and periodontitis as a manifestation of systemic disease, which were not included in this NMA. These category differences explain the dissimilar outcomes between the previous and present NMA.

When evaluating the clinical significance of AMOX+MET as an adjunct to subgingival debridement, the combination of these antibiotics showed no observable clinical effect on CAL gain or PPD reduction at all follow-up periods. This finding is in concordance with previous studies [28,93] where no observable clinical effect was detected for CAL gain or PPD reduction. In terms of BOP reduction, AMOX+MET showed MICD regardless of the follow-up period. However, these findings were of low to very low certainty, except for long-term BOP reduction which had high certainty of evidence. The low certainty of evidence is contributed to mainly by serious risk of bias, inconsistency, indirectness, and imprecision within the studies that embodied the evidence. Therefore, the findings should be interpreted with caution. Future well-designed and high-quality RCTs are required to determine the true adjunctive effect of AMOX+MET in the treatment of periodontitis.

With regards to the relative effect of AZ as an adjunct to subgingival debridement in the treatment of patients with periodontitis, the results showed that trivial to no clinical difference was observed following its adjunctive use for all outcomes at any point of time. Moreover, AZ was found to be less effective than subgingival debridement for all long-term outcomes. However, the evidence is of very low certainty due to serious risk of bias, inconsistency, indirectness, and imprecision within the pooled studies. Thus, this finding should be interpreted with caution. Well-designed RCTs are required to enable a strong conclusion with regards to the adjunctive effect of this antibiotic. When compared with previous NMA by Sgolastra et al. [28], both studies reported similar findings in which the authors found no significant difference for all outcomes at any time-point following the adjunctive use of AZ in the treatment of patients with chronic periodontitis. Therefore, based on the findings of the previous and present NMA, it can be concluded that AZ as an adjunct to subgingival debridement may not be effective for treating periodontitis, and the use of AZ in current clinical practice may need reconsideration.

### 4.3. Strengths and Limitations

To the best of our knowledge, this is the first NMA to assess the effectiveness of commercially available systemic antibiotics adjunctive to subgingival debridement in the treatment of patients with periodontitis, along with evaluation of the evidence certainty using a GRADE approach. This approach was adopted to provide evidence that will be useful and comprehensible for clinicians and decision-makers. Moreover, this NMA considered the potential confounding effect that other active adjuncts (e.g., chlorhexidine) may have on the effect estimates when used in combination with the systemic antibiotics.

In the conduct of the NMA certain limitations were identified, as follows:There was substantial (I^2^ > 50%) between-studies heterogeneity in most of the network estimates. To overcome this limitation, subgroup analyses were conducted by splitting the studies into less heterogenous groups and separate analyses were performed for each group to explore potential reasons for differences.Numerous studies included parameters (outliers and influential cases, imputed/transformed missing data, and high risk of bias) that may have affected the estimates of treatment effect. To address this limitation, sensitivity analyses were conducted by re-running the NMA after excluding studies with such parameters, to ensure the robustness of the treatment-effect estimates.More than half of the included studies were at high risk of bias. In addition to removing such studies during the sensitivity analysis, the certainty of evidence provided by studies with a high risk of bias was downgraded to avoid misleading interpretation of the network estimates.

## 5. Conclusions

Evidence with very low to high certainty is available to inform the effectiveness of systemic antibiotics adjunctive to subgingival debridement in the treatment of patients with periodontitis. Clinical outcomes ranging from minimally important clinical difference to trivial to no clinical difference were observed following the adjunctive use of these antibiotics. The adjunctive use of these antibiotics should be weighed against possible harm to avoid their overuse and inappropriate use in patients with periodontitis.

## Figures and Tables

**Figure 1 antibiotics-11-01716-f001:**
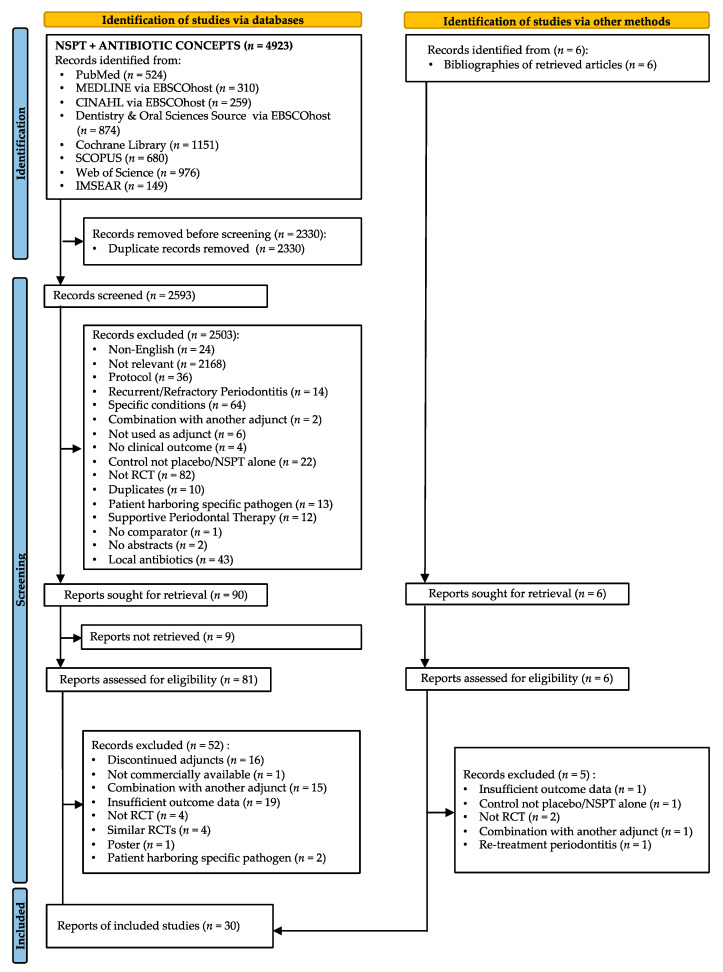
Preferred reporting items for systematic reviews and meta-analyses (PRISMA) flow diagram of the study-selection process.

**Figure 2 antibiotics-11-01716-f002:**
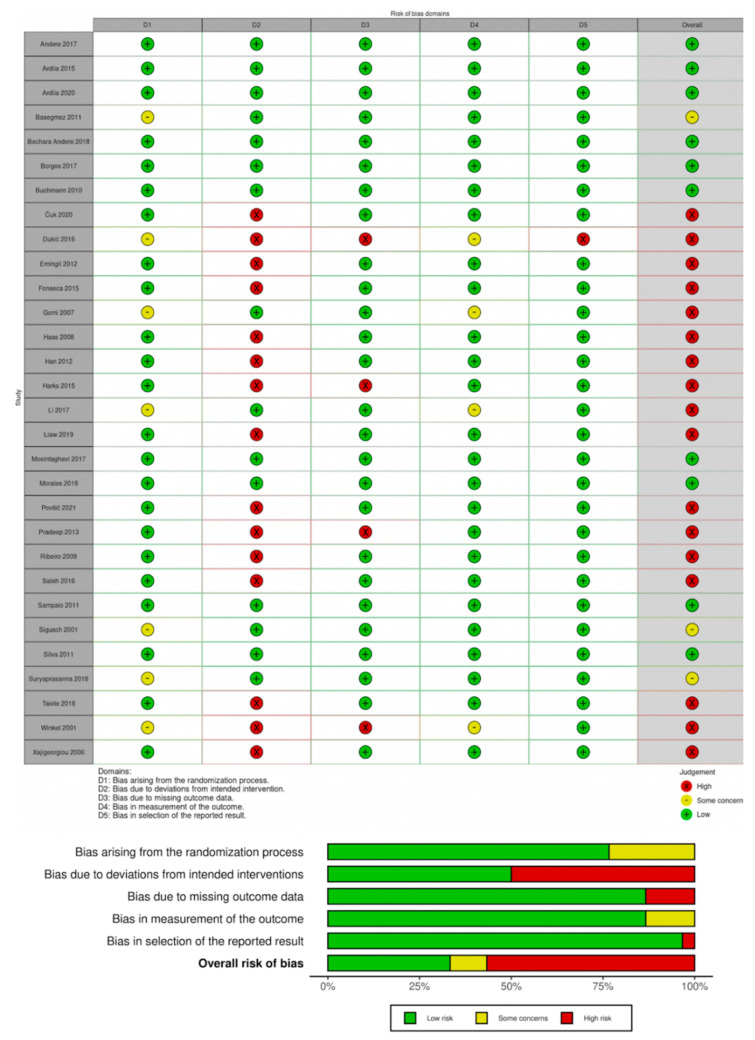
Risk-of-bias summary of the included studies. Andere 2017 [61]; Ardila 2015 [63]; Ardila 2020 [62]; Basegmez 2011 [53]; Bechara Andere 2018 [64]; Borges 2017 [65]; Buchmann 2010 [73]; Čuk 2020 [74]; Dukić 2016 [75]; Emingil 2012 [54]; Fonseca 2015 [66]; Gomi 2007 [55]; Haas 2008 [67]; Han 2012 [56]; Harks 2015 [76]; Li 2017 [57]; Liaw 2019 [81]; Moeintaghavi 2017 [58]; Morales 2016 [68]; Povšič 2021 [77]; Pradeep 2013 [59]; Ribeiro 2009 [69]; Saleh 2016 [82]; Sampaio 2011 [70]; Sigusch 2001 [78]; Silva 2011 [71]; Suryaprasanna 2018 [60]; Taiete 2016 [72]; Winkel 2001 [79]; Xajigeorgiou 2006 [80].

**Figure 3 antibiotics-11-01716-f003:**
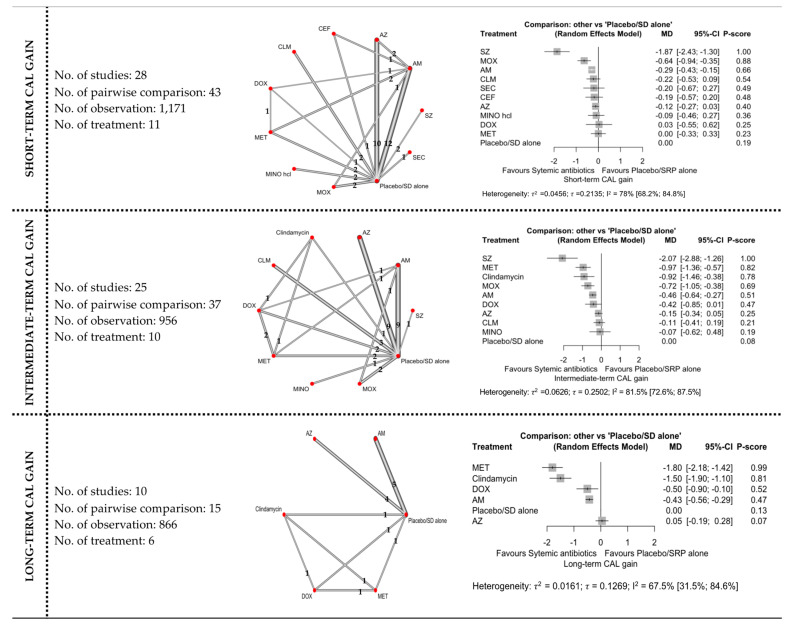
The network and forest plots of the network meta-analysis for CAL gain. AM: amoxycillin + metronidazole; AZ: azithromycin; CEF: cefixime; CLM: clarithromycin; DOX: doxycycline; MET: metronidazole; MINO: minocycline; MINO hcl: minocycline hydrochloride; MOX: moxifloxacin; SEC: secnidazole; SZ: satranidazole; SD: subgingival debridement.

**Figure 4 antibiotics-11-01716-f004:**
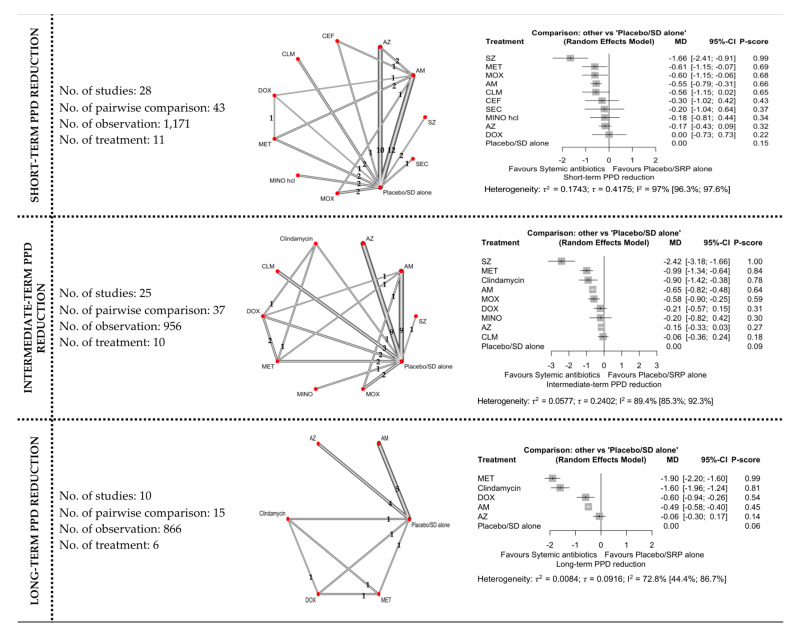
The network and forest plots of the network meta-analysis for PPD reduction. AM: amoxycillin + metronidazole; AZ: azithromycin; CEF: cefixime; CLM: clarithromycin; DOX: doxycycline; MET: metronidazole; MINO: minocycline; MINO hcl: minocycline hydrochloride; MOX: moxifloxacin; SEC: secnidazole; SZ: satranidazole; SD: subgingival debridement.

**Figure 5 antibiotics-11-01716-f005:**
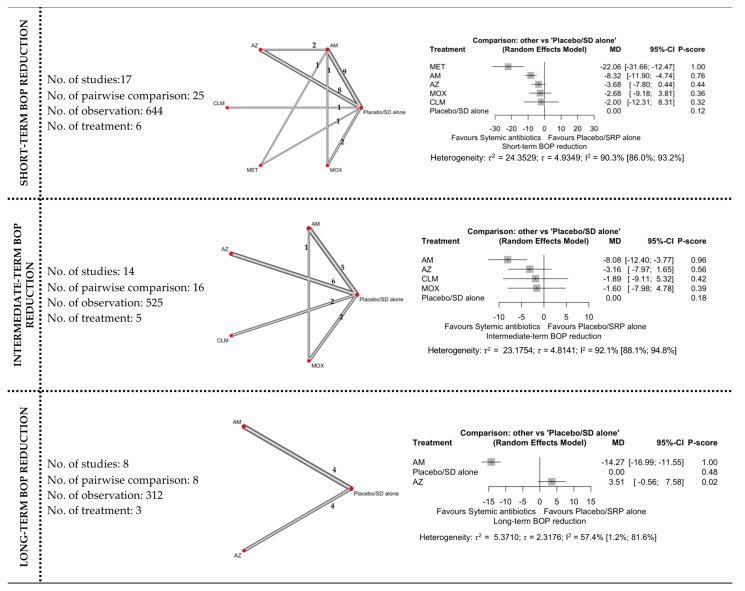
The network and forest plots of the network meta-analysis for BOP reduction. AM: amoxycillin + metronidazole; AZ: azithromycin; CEF: cefixime; CLM: clarithromycin; DOX: doxycycline; MET: metronidazole; MINO: minocycline; MINO hcl: minocycline hydrochloride; MOX: moxifloxacin; SEC: secnidazole; SZ: satranidazole; SD: subgingival debridement.

**Figure 6 antibiotics-11-01716-f006:**
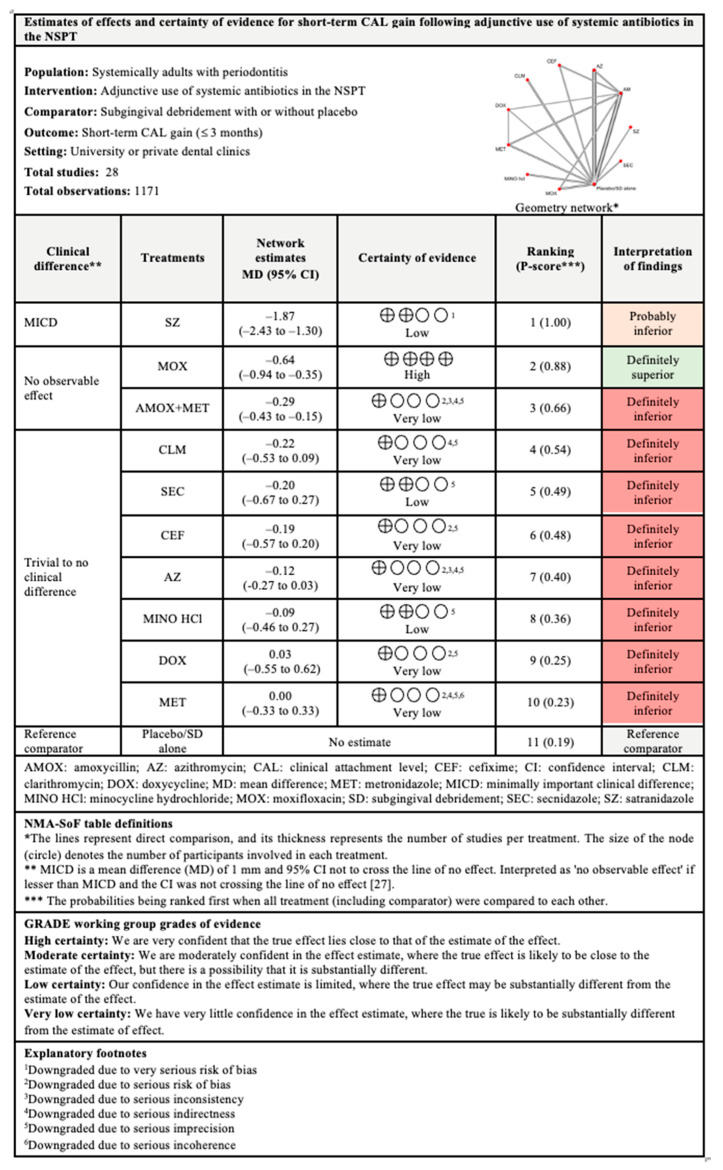
Summary of findings for short-term CAL gain following the use of systemic antibiotics adjunctive to subgingival debridement.

**Figure 7 antibiotics-11-01716-f007:**
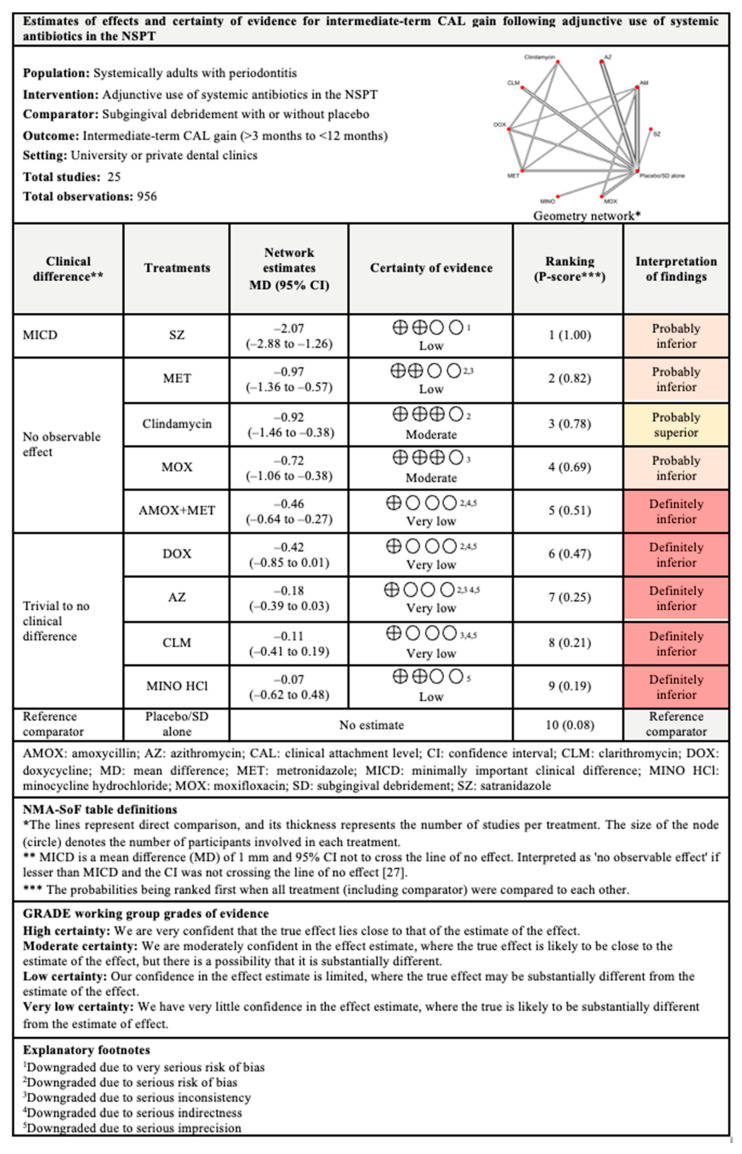
Summary of findings for intermediate-term CAL gain following the use of systemic antibiotics adjunctive to subgingival debridement.

**Figure 8 antibiotics-11-01716-f008:**
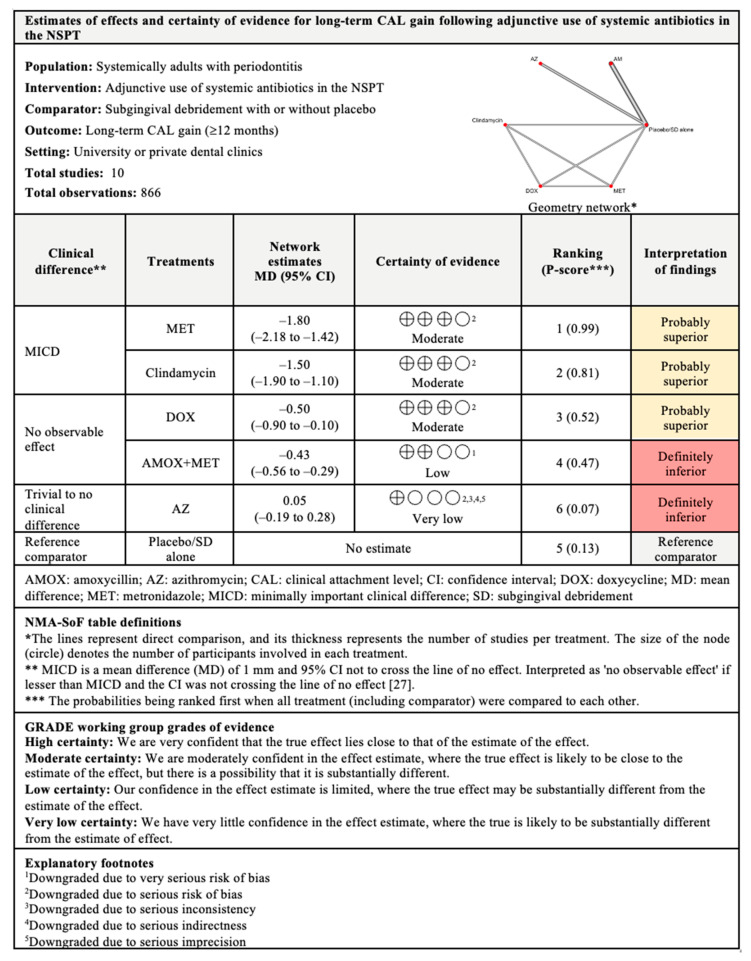
Summary of findings for long-term CAL gain following the use of systemic antibiotics adjunctive to subgingival debridement.

**Figure 9 antibiotics-11-01716-f009:**
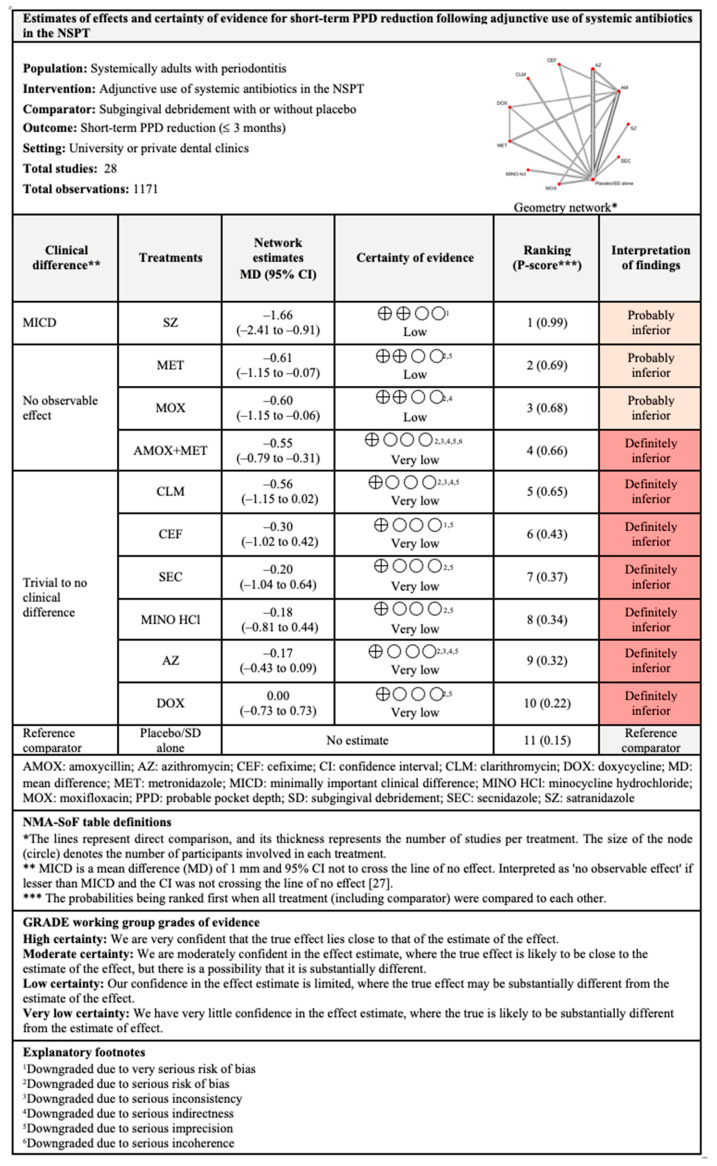
Summary of findings for short-term PPD reduction following the use of systemic antibiotics adjunctive to subgingival debridement.

**Figure 10 antibiotics-11-01716-f010:**
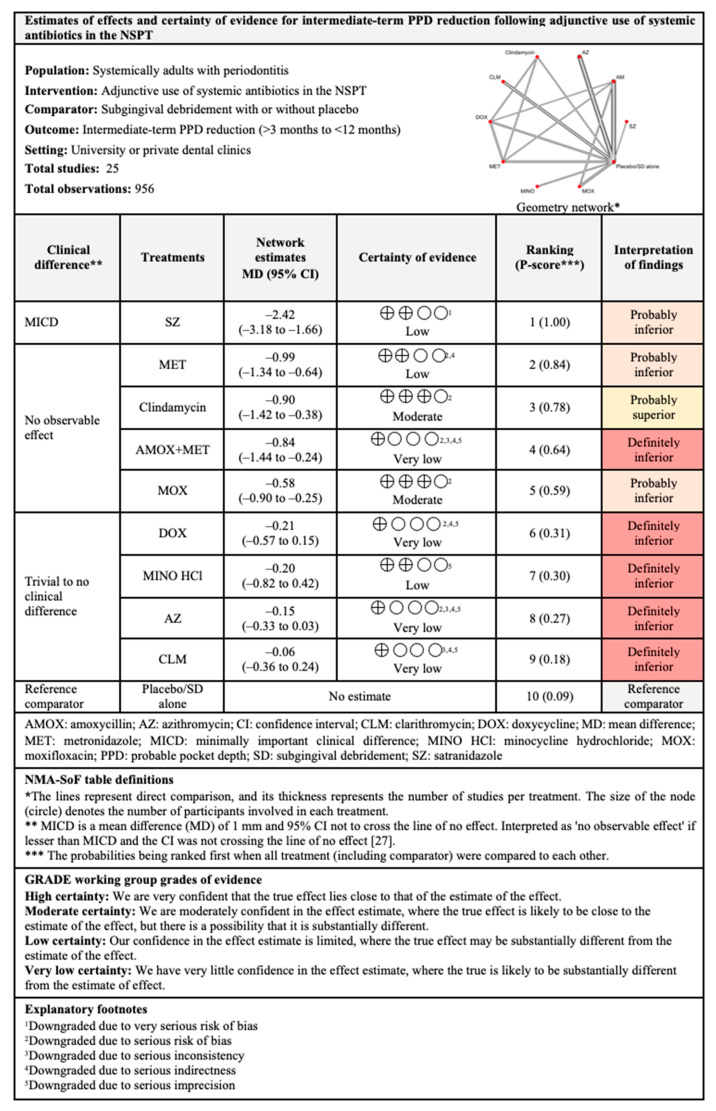
Summary of findings for intermediate-term PPD reduction following the use of systemic antibiotics adjunctive to subgingival debridement.

**Figure 11 antibiotics-11-01716-f011:**
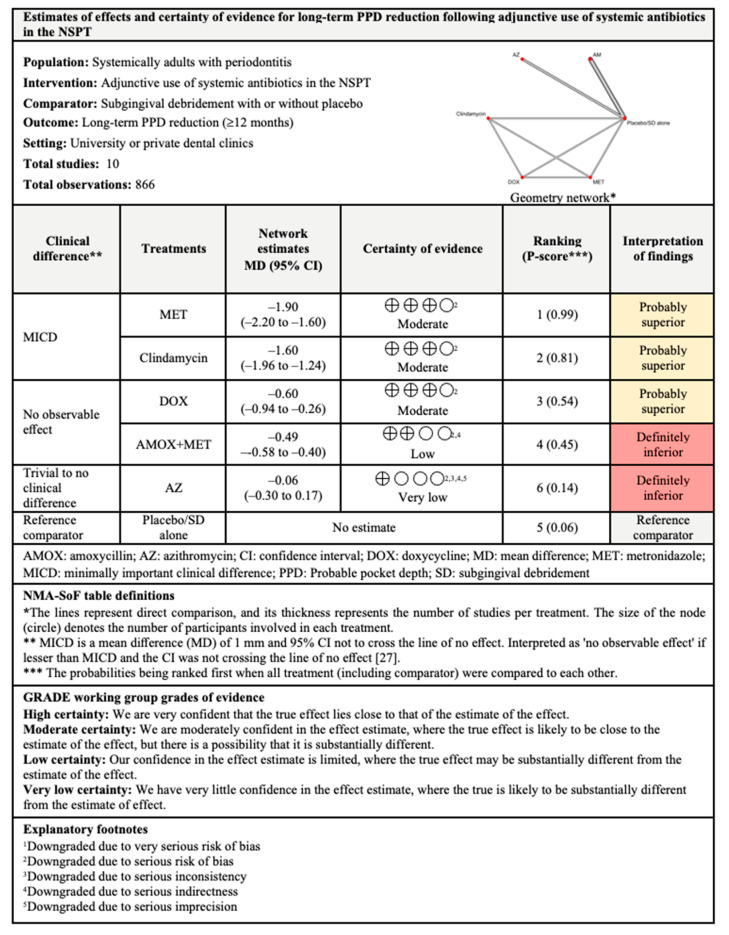
Summary of findings for long-term PPD reduction following the use of systemic antibiotics adjunctive to subgingival debridement.

**Figure 12 antibiotics-11-01716-f012:**
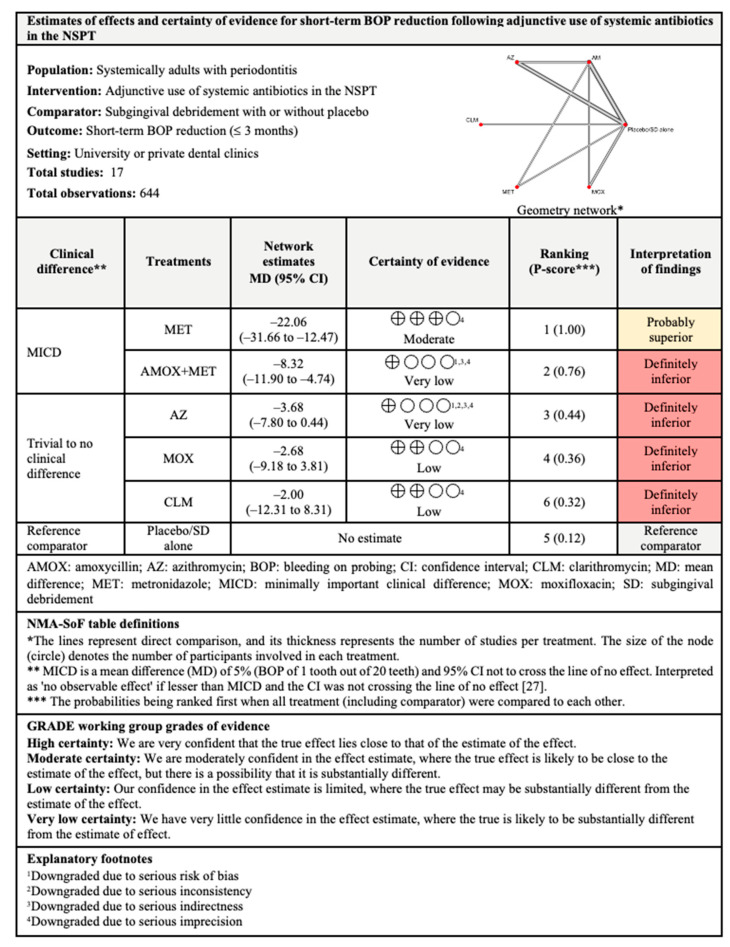
Summary of findings for short-term BOP reduction following the use of systemic antibiotics adjunctive to subgingival debridement.

**Figure 13 antibiotics-11-01716-f013:**
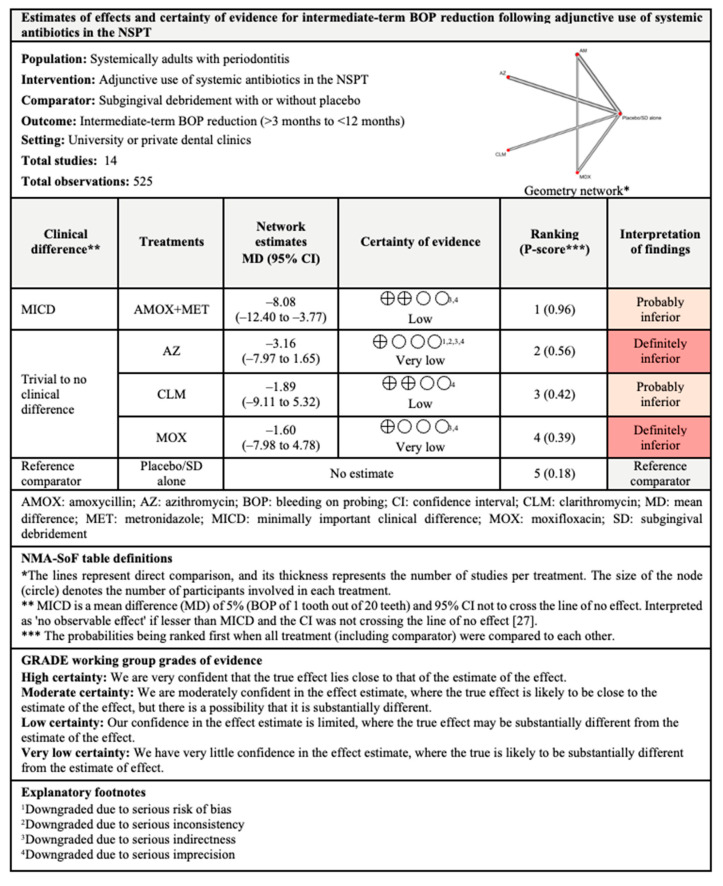
Summary of findings for intermediate-term BOP reduction following the use of systemic antibiotics adjunctive to subgingival debridement.

**Figure 14 antibiotics-11-01716-f014:**
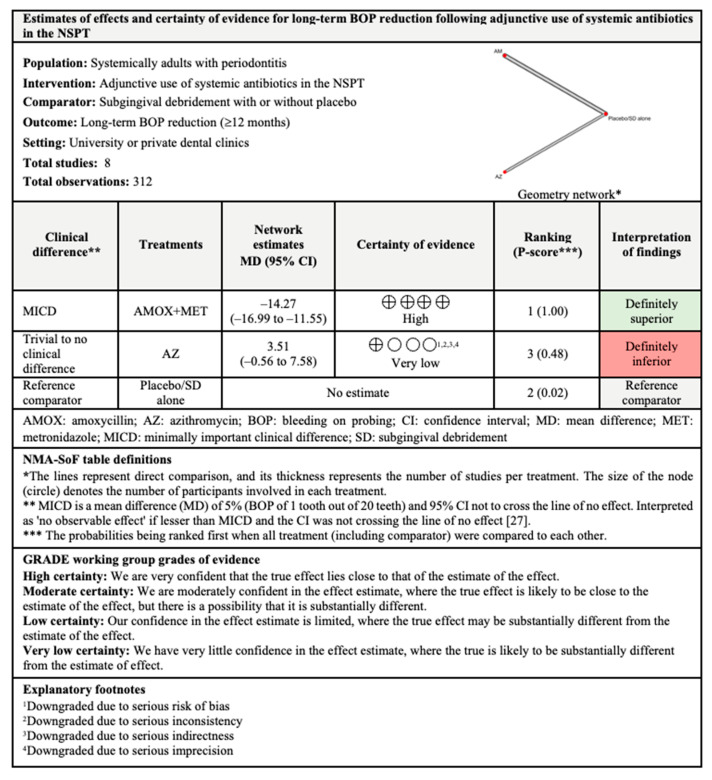
Summary of findings for long-term BOP reduction following the use of systemic antibiotics adjunctive to subgingival debridement.

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
