# Peer review of "Systemic Antibiotics as an Adjunct to Subgingival Debridement: A Network Meta-Analysis"

_antibiotics, 2022, doi:10.3390/antibiotics11121716_

Round 1

Reviewer 1 Report

This systematic review is an excellent approximation to a network meta-analysis to assess the effectiveness of systemic antibiotics currently marketed as an adjunct to periodontal therapy. The methodology meets all the criteria of a systematic network review following the GRADE recommendations. However, due to multiple reasons stated by the authors, such as the high-risk bias and high heterogeneity, it was not possible to establish the effectiveness of the use of antibiotics in periodontal treatment with a high degree of certainty.

Below are some respectful suggestions to the authors.

1)      AMX-MTZ has been the most effective treatment in previous pairwise and other NMA reviews and is considered the standard treatment. This effect was not observed for the most critical variables, such as CAL and PPD, in this review. The authors argue that this effect could be due to its combination with CHX. In order to confirm this result, it is required that the authors use some strategy to evaluate in a more objective way than the use of CHX if it was such a relevant variable. It is very relevant to demonstrate this hypothesis with statistical results. Exclusion for this reason alone is not as convincing.

2)      The authors state in the introduction, "Nevertheless, the authors did not recommend routine use of systemic antibiotics as an adjunct but may be considered when treating specific periodontal conditions such as generalized Stage III periodontitis in young adults". An analysis of subgroups by type of periodontitis is indicated in the methodology. However, this is not considered in the discussion and is highly relevant to establish differences.

3)      Support how the analysis of subgroups could influence the results of the NMA.

4)      satranidazole (SZ) is not marketed in all countries and the quality of the evidence is low and there are only short-term results. Better support its clinical use and adverse effects in the discussion.

5)      Present with greater clarity relevant methodological aspects such as the analysis of homogeneity, assumption of consistency, and certainty of the evidence within StrengthsThis systematic review is an excellent approximation to a network meta-analysis to assess the effectiveness of systemic antibiotics currently marketed as an adjunct to periodontal therapy. The methodology meets all the criteria of a systematic network review following the GRADE recommendations. However, due to multiple reasons stated by the authors, such as the high-risk bias and high heterogeneity, it was not possible to establish the effectiveness of the use of antibiotics in periodontal treatment with a high degree of certainty.

Author Response

Dear Reviewer,

Thank you for your kind feedback and constructive input. Please see the attachment for the responses to your review. Your kind review and time spent is much appreciated. Thank you.

Reviewer 2 Report

Thank you very much for the article titled “Systemic Antibiotics as Adjunct to Subgingival Debridement: A Network Meta-Analysis”

This paper is a systematic review addressing evaluate the effectiveness of systemic antibiotics as an adjunct to subgingival debridement in the treatment of patients with periodontal disease. This is a topic which merits discussion and has clinical relevance for the audience of the journal. The methodology closely follows Cochrane methodology and, as such, is comprehensive and properly conducted.

The authors described systemic antibiotics used in periodontitis patients, but in my opinion, the results should of the analysis performed by the authors should be interpreted with caution. In the text, I couldn’t find the information- do current guidelines for using specific antibiotics in such patients? Are guidelines adequate? Do they need to be more prescriptive? Maybe this information should be added in the Discussion section.

Author Response

(The authors gave the same response as above.)

Reviewer 3 Report

This paper aims to evaluate the effectiveness of systemic antibiotics as adjunct to sub- gingival debridement in the treatment of patients with periodontitis in a meta-analysis protocol. It is a relevant study, in a well-structured and organized document. The theme is not innovative but rigorous impressions were made and can add relevance to the scientific evolution of the periodontal treatment.

Abstract: Acronyms of CAL, PPD and BOP should be described in extension in the abstract or not used at all, to facilitate clear comprehension of the abstract and invite readers to read the full text. In my point a view, the abstract should not contain abbreviations at all.

Introduction – should include the possible adverse effects of systematic therapy with drugs. Include adverse effects like a problem of systematic therapy and not only microbial resistance. I think resistance should be mentioned as a global problem for all bacterial (not only the responsible for oral diseases).

In this chapter seems that antibiotics are used only on class III periodontal disease. I advise to mentioned that literature include other types of the disease.

In this chapter and through the document it should be very clear that ONLY studies with NON SURGICAL periodontal treatment were selected.

Material and Methods:

Variables integrated in data extraction and subgroup analysis should be described in this chapter (e.g. type of periodontitis- describe what types where used, follow-up period- describe what means short term, intermediate term and long term???

Pilot studies (included in the eligibility factors) should not be included in this level of evidence when the number of RCT are evidently sufficient. Why did the authors include them?

Non-surgical studies should be outlined in the intervention description.

Results- examples of adverse effects should be included in the text.

Intervention characteristics subtitle- included various therapeutic drugs but strangely did not include clindamycin, which was described in the abstract as one of three principal drugs that promotes CAL gain and PPD reduction. What is the real importance of this antibiotic? How many studies used it in adjuvant to mechanical therapy?

Author Response

(The authors gave the same response as above.)

Reviewer 4 Report

This meta-analysis is  very well organized and innovative due the network meta-analysis model. The results are very useful for the clinical periodontist and the discussion very well structured.

Author Response

Dear Reviewer,

Thank you for your kind feedback. Your time spent is much appreciated. Thank you.

Round 2

Reviewer 1 Report

Now, this article can be published in the Antibiotics journal